Optimizing nutrient management protocol for Ophiopogon japonicus-corn intercropping: impacts on growth, yield, and medicinal quality

Cai Xiaoyang
Fan Heling
Deng Hongmei
Li Wenjing
Wang Haohan
Zhang Jiaming
Li Min 028limin@163.com
1 Chengdu University of Traditional Chinese Medicine , Chengdu , China
2 Sichuan Research Center for Demonstration Project of Entire Industrial Chain of Genuine Medicinal Materials , Chengdu , China
Fayed Marwa
Electronic publication date: 2025 Jul 14
Publication date: 2025
Volume: 13
Electronic Location ID: e19655
Received 2024 Dec 11; Accepted 2025 Jun 3
Copyright: ©2025 Cai et al.
Copyright year: 2025
Copyright holder: Cai et al.
License: This is an open access article distributed under the terms of the Creative Commons Attribution License, which permits unrestricted use, distribution, reproduction and adaptation in any medium and for any purpose provided that it is properly attributed. For attribution, the original author(s), title, publication source (PeerJ) and either DOI or URL of the article must be cited.
License URL: https://creativecommons.org/licenses/by/4.0/

Keywords: Intercropping system, Nitrogen fertilizer types, Micronutrients, Top-dressing, Yield and quality

Funding: Study on Ecological Regulation Cultivation Technology for High-Quality Traditional Chinese Medicine 2022YFC3501500 This research was funded by the Study on Ecological Regulation Cultivation Technology for High-Quality Traditional Chinese Medicine, grant number 2022YFC3501500. The funders had no role in study design, data collection and analysis, decision to publish, or preparation of the manuscript.

==============================
Ophiopogon japonicus is a valuable medicinal plant commonly intercropped with corn due to limited arable land in its primary cultivation areas. Optimizing fertilization management to enhance both crop yields and medicinal quality is a key research focus. This study investigates the effects of different fertilization treatments—nitrogen sources (urea and ammonium nitrate), micronutrient supplementation (magnesium and manganese), and phosphorus-potassium (P+K) fertilization—on the growth, yield, and medicinal quality of O. japonicus and corn in an intercropping system. A randomized complete block design was employed to evaluate eight treatments. Agronomic traits, biomass production, and medicinal quality indicators were analyzed using cluster analysis, correlation analysis, and principal component analysis (PCA). Results showed that nitrogen source significantly affected corn height, with ammonium nitrate outperforming urea. Micronutrients and P+K had significant interactive effects on stem diameter and leaf development. Fresh biomass and silage yield were primarily influenced by nitrogen source and P+K, while stem biomass was affected by micronutrient supplementation. The highest corn yield and biomass were recorded in treatments T5 (urea + P + K), T7 (urea + Mn + Mg + P + K), and T8 (ammonium nitrate + Mn + Mg + P + K). For O. japonicus, micronutrient applications significantly increased tuberous root numbers, while P+K fertilization promoted plant height and fibrous root growth. All three fertilization factors significantly impacted tuber biomass, the main determinant of medicinal yield. T8 showed the highest overall yield of O. japonicus, followed by T5 and T7. Medicinal quality evaluation through cluster analysis and PCA identified T5 as the optimal treatment for enhancing key medicinal components. The optimal strategy for maximizing yield in the intercropping system includes ammonium nitrate (150 kg N/ha), magnesium sulfate (45 kg/ha), manganese sulfate (15 kg/ha), superphosphate (75 kg P2O5/ha), and potassium sulfate (450 kg K2O/ha). For improving medicinal quality, the best treatment includes urea (150 kg N/ha), superphosphate (75 kg P2O5/ha), and potassium sulfate (450 kg K2O/ha). Future studies should assess the adaptability of this intercropping system across different soil and climatic conditions. Incorporating precision agriculture technologies may further refine fertilization strategies, while long-term monitoring is recommended to evaluate impacts on soil health and environmental sustainability.

Introduction

With the rising global health awareness and shifts in medical paradigms, the efficacy and safety of natural medicines are increasingly recognized in the global market. Traditional Chinese medicine, as a significant component of natural medicine, has also gained widespread acknowledgment for its effectiveness and safety. Modern research has identified the main chemical constituents of Ophiopogon japonicus, including steroid saponins (Sun et al., 2013; Wang et al., 2011), high isoflavonoids (Lin, Liu & Ye, 2014; Zhou et al., 2013), and polysaccharides (Fang et al., 2018; Xiong et al., 2011), which exhibit various pharmacological effects such as lowering blood sugar (Ding et al., 2012; Wang, 2013), protecting the cardiovascular system (Lan et al., 2013; Zheng et al., 2009), enhancing immunity (Xiong et al., 2011), anti-aging (Xiong et al., 2011), anti-inflammatory (Song et al., 2010), and anti-tumor activities (Chen et al., 2013; Zhang et al., 2012). Due to its efficacy and safety (Jin et al., 2024), Ophiopogon japonicus and its products are widely used in at least 25 countries, including China, Japan, Germany, Vietnam, India, the United States, Malaysia, and South Korea. According to customs data, China exported over 6,000 tons of Ophiopogon japonicus from 2018 to 2022. Santai County in Sichuan Province is a major production area for Ophiopogon japonicus with a long cultivation history, yielding approximately 15,000 tons annually, which accounts for over 90% of the domestic market. The development of regional planting and specialized production has made Ophiopogon japonicus a leading industry in local rural areas and a significant source of income for farmers. However, the suitable land for Ophiopogon japonicus cultivation is limited, compounded by policies aimed at preventing the “non-food” use of arable land (Dong et al., 2024; Li et al., 2024a; Su et al., 2024), leading to competition for space between medicinal and food crops. Additionally, Ophiopogon japonicus cultivation faces challenges such as soil pH imbalance, pollution, and biodiversity loss (Bertness & Callaway, 1994; Moonen & Bàrberi, 2008).

Based on local Ophiopogon japonicus production practices and policy guidance, a sustainable “corn–Ophiopogon japonicus” intercropping system has been developed by integrating traditional agriculture with modern technology. This ecological planting method now accounts for over 90% of the cultivated area in production regions and plays a crucial role in Ophiopogon japonicus farming. First, intercropping enhances biodiversity within agricultural systems. Studies show that intercropping with legumes increases the abundance of beneficial bacteria; for example, intercropping with paulownia and buckwheat significantly improves soil bacterial diversity. A richer microbial community and higher enzyme activity contribute to nutrient cycling and soil health, particularly by increasing soil organic matter, nitrogen, and potassium levels (Simon, Mmateko & Ochanda, 2024; Woźniak et al., 2025; Zhou et al., 2024). Second, this intensive land-use approach greatly improves land utilization efficiency. Intercropping systems, such as legumes with oilseeds (e.g., pea–canola and chickpea–flax) (McAuley, Bourgault & Congreves, 2025) and legumes with corn (Wang et al., 2024a), have demonstrated higher land equivalent ratios. Additionally, intercropping promotes crop trait plasticity, which helps stabilize productivity over time (Yang et al., 2025). Third, intercropping improves crop quality. For instance, intercropping Aconitum carmichaeli with local crops significantly enhances its quality by increasing biomass and polysaccharide content while altering alkaloid accumulation (Liu et al., 2023). Similarly, tobacco–Isatis intercropping improves tobacco leaf chemical composition (Wang et al., 2024b), and intercropping Eugenia dysenterica enhances fruit nutritional quality, increasing nitrogen levels, essential minerals, phenolic content, and antioxidant capacity (Almeida et al., 2024). Furthermore, intercropping has been shown to boost flavonoid and saponin levels in carrots by 32.92% and 13.92%, respectively (Zhou et al., 2024). Finally, intercropping increases per-unit land revenue, significantly improving farmers’ economic returns. Systems such as rice–corn and legume–corn intercropping have been shown to enhance profitability (Li et al., 2024c; Metwally, Abdel-Wahab & Abdel-Wahab, 2019; Riyanto, Anshori & Srihartanto, 2021). The adoption of this ecological planting model fosters green development while simultaneously promoting food and medicinal plant production and increasing farmers’ income.

Intercropping influences crop growth and yield composition through spatial and temporal arrangements (Yang et al., 2015). By leveraging complementary growth patterns among different species, intercropping optimizes resource utilization, leading to various benefits, including increased productivity, improved soil health, and effective weed management (Tong et al., 2024; Villegas-Fernández et al., 2024). However, due to differences in crop structures, intercropping can result in both nutrient facilitation and competition (Davis & Liebman, 2001; Ghosh et al., 2009). Enhanced rhizosphere interactions and microbial community dynamics in intercropping systems significantly improve nutrient uptake. For instance, tobacco–corn intercropping increases nitrogen (N), phosphorus (P), and potassium (K) accumulation in both crops (Dou et al., 2025). Similarly, spring wheat-pea intercropping raises N and P absorption rates by 11.4% and 11.3%, respectively (Feng et al., 2024). In soybean-corn intercropping, specific microorganisms such as Microbacterium and Rhizobium produce iron chelators that facilitate nutrient absorption under iron-deficient conditions (Liu et al., 2025). However, intercropping can also intensify nutrient competition, potentially affecting overall productivity and resource efficiency. For example, Suaeda salsa-corn intercropping creates competition for N, negatively impacting corn growth, though nitrogen fertilization can mitigate this effect (Wang et al., 2022). Under nutrient-limited conditions, competition between millet and Guinea grass significantly influences nutrient uptake and growth (Jayamanna et al., 2023). Additionally, intercropping alters the uptake and mobility of micronutrients such as manganese (Mn) and magnesium (Mg). Corn-soybean intercropping has been shown to enhance the bioavailability of Mg and other micronutrients, particularly when combined with appropriate fertilizers (Dragicevic et al., 2015). Conversely, corn intercropped with Sedum alfredii exhibited reduced Mn absorption, whereas Mn levels in soybean remained unchanged (Li et al., 2024d). In practice, improper fertilization techniques and mismanagement can lead to low nutrient use efficiency and significant nutrient losses, posing risks to agricultural productivity, resource conservation, and environmental sustainability. Existing fertilization research on Ophiopogon japonicus primarily focuses on monocropping, with no studies addressing nutrient supplementation in the “corn–Ophiopogon japonicus” intercropping system. This knowledge gap results in uncoordinated fertilization strategies, leading to suboptimal nutrient ratios and inefficient application. Given that corn and Ophiopogon japonicus are a high-low crop combination, their canopy structure facilitates greater light interception, enhancing overall light-use efficiency and yield potential (Awal, Koshi & Ikeda, 2006). However, optimizing nutrient supplementation in this multi-cropping system remains a critical challenge. Previous studies indicate that both corn and Ophiopogon japonicus require substantial nitrogen inputs, with corn, a Poaceae crop, exhibiting a competitive advantage in N uptake. This further influences the nitrogen use efficiency of Ophiopogon japonicus (Li et al., 2001). Currently, fertilization practices primarily target Ophiopogon japonicus, failing to meet the nutrient demands of both crops, especially for nitrogen. Urea and ammonium nitrate are the primary nitrogen fertilizers used in production areas, yet different nitrogen forms affect plant physiology and root development differently. Determining the most suitable nitrogen fertilizer for the “corn-Ophiopogon japonicus” intercropping system requires further investigation. In addition to nitrogen, micronutrients such as Mg and Mn should also be considered. Studies suggest that supplementing these elements not only alleviates nutrient competition but also ensures the medicinal quality of Ophiopogon japonicus (Aulakh & Malhi, 2005).

Currently, there is limited research on nutrient management protocols in the “Ophiopogon japonicus-corn” multi-cropping system. This study will focus on three key factors: nitrogen source selection, N-P-K ratio optimization, and micronutrient supplementation. The objective is to investigate their effects on the growth of corn and Ophiopogon japonicus, as well as the medicinal quality of Ophiopogonis Radix (the medicinal part of Ophiopogon japonicus refers to its dried root tubers). The findings will provide a scientific basis for nutrient management in the “Ophiopogon japonicus-corn” multi-cropping system, ensuring high yield, stable production, and superior quality of Ophiopogonis Radix.

Material & Methods

Experimental design

This study was conducted at the Ophiopogon japonicus Research Demonstration Park in Santai County, Sichuan Province (Longitude 104°57′44″, Latitude 31°24′35″). The area has a subtropical monsoon climate with humid conditions and sandy loam soil. Key soil characteristics include a pH of 7.05, organic matter at 10.3 g/kg, ammonium nitrogen at 28.2 mg/kg, available phosphorus at 81.2 mg/kg, and available potassium at 124.8 mg/kg. The research aims to assess the effects of various nutrient management protocols on the growth, yield, and medicinal quality of Ophiopogon japonicus and corn in an intercropping system (Fig. 1).

Figure 1 Intercropping diagram of Ophiopogon japonicus and corn.

Uniform basal fertilizer management was implemented across all treatments, utilizing 1.5 t/hm2 of commercial organic fertilizer (N+P2O5+K2O ≥ 5%, organic matter ≥ 45%, produced by Mianyang Keya Agricultural Development Co., Ltd.) and 600 kg/hm2 of compound fertilizer (N: P2O5: K2O = 17: 17: 17, total nutrients ≥ 51%, produced by Guizhou Xiyang Industry Co., Ltd.).

The nutrient management protocols experiment employed a complete randomized block design with three factors and two levels, resulting in eight treatments, each replicated three times across 24 plots, each measuring 24 m2. The first factor examined types of nitrogen fertilizers, comparing urea (N ≥ 46%) and ammonium nitrate (N ≥ 34.8%), applied at 150 kg N/hm2. The second factor focused on micronutrient fertilizers, specifically magnesium and manganese, with treatments for magnesium sulfate (MgSO4 ⋅ 7H2O) applied at 45 kg/hm2 and manganese sulfate (MnSO4 ⋅ H2O) at 15 kg/hm2, both as foliar applications. The third factor evaluated topdressing with phosphorus and potassium fertilizers, applying Calcium superphosphate (P2O5 ≥ 12%) at 75 kg P2O5/hm2 and Potassium sulfate (K2O ≥ 52%) at 450 kg K2O/hm2 (Table 1).

Corn was planted at a density of 45,000 plants/hm2, with ten rows of Ophiopogon japonicus interspersed between two rows of corn, resulting in a planting density of 1.5 million plants/hm2. The Ophiopogon japonicus variety used was “Chuanmaidong No. 1”, the local cultivar, while the corn variety was “Chengdan 99”, intended for silage production. Both Ophiopogon japonicus and corn were sown on April 22, 2022, with management practices following the standard (China Association of Chinese Medicine, 2020). The corn was harvested on July 26, 2022, and the Ophiopogon japonicus on March 25, 2023.

Methods for determination of agronomic indicators

In the experimental design, both Ophiopogon japonicus and corn will be harvested at their respective maturity stages. Corn will be harvested by removing the entire aboveground portion, while Ophiopogon japonicus will be excavated from the soil to obtain the entire plant. The height of Ophiopogon japonicus and corn was measured using a measuring tape, while the stem circumference of corn was measured with a flexible tape. The number of corn leaves was recorded, excluding those with over 50% wilting. The quantities of storage roots, nutritive roots, and root tubers of Ophiopogon japonicus were noted. Fresh weights of various parts—corn (including corn with husks, leaves, and stems) and Ophiopogon japonicus (leaves, rhizomes, storage roots, nutritive roots, and root tubers)—were measured using a 0.01 g electronic balance. As shown in Fig. 2, the fresh weight of the root tuber represents the yield of Ophiopogon japonicus. The medicinal material of Ophiopogon japonicus was graded according to standards (Mianyang Market Supervision and Administration Bureau China, 2023).

Table 1 Nutrient management protocols design table for the Ophiopogon japonicus-corn intercropping system.

Treatment code	Types of nitrogen fertilizers	Micronutrient fertilizers	Topdressing	
T1	Urea	/	/	
T2	Ammonium nitrate	/	/	
T3	Urea	Mn+Mg	/	
T4	Ammonium nitrate	Mn+Mg	/	
T5	Urea	/	P+K	
T6	Ammonium nitrate	/	P+K	
T7	Urea	Mn+Mg	P+K	
T8	Ammonium nitrate	Mn+Mg	P+K	

Figure 2 Diagram of Ophiopogon japonicus parts.

Methods for determining quality indicators of Ophiopogon japonicus medicinal material

The determination of moisture, extractives, and total saponins for Ophiopogon japonicus was conducted according to the methods outlined in the 2020 edition of the Pharmacopoeia of China (Chinese Pharmacopoeia Commission, 2020). The total flavonoid content in the medicinal material was measured using ultraviolet spectrophotometry, with hesperidin as the reference standard (Lin, 2014). The total polysaccharide content was determined using the sulfuric acid”-anthrone colorimetric method (Wang et al., 2016).

Standards for Ophiopogonin D (reference substance number Wkg22061609), methylophiopogonanone A (reference substance number Wkq24021807), and methylophiopogonanone B (reference substance number Wkq23072006) were purchased from Sichuan Weikegi Biological Technology Co., Ltd. (Chengdu, China). High-performance liquid chromatography (HPLC)-grade acetonitrile was obtained from Sichuan Cologne Chemical Co., Ltd. (Chengdu, China). HPLC was employed to determine the content of Ophiopogonin D, methylophiopogonanone A, and methylophiopogonanone B in the samples.

For the extraction of Ophiopogonin D, three g of Ophiopogon japonicus powdered material was placed in a round-bottom flask with 50 mL of methanol. The mixture was refluxed for 2 h, filtered, and the filtrate was concentrated to dryness. Afterward, 10 mL of water was added to dissolve the residue. The solution was then extracted with water-saturated n-butanol, shaking five times with 12 mL each time. The n-butanol extracts were combined and washed twice with five mL of ammonium hydroxide solution. The ammonium solution was discarded, and the n-butanol solution was evaporated to dryness. Finally, the residue was dissolved in 80% methanol, transferred to a five mL volumetric flask, and filtered through a 0.45 µm membrane filter (Wu et al., 2015).

The chromatographic conditions for Ophiopogonin D included using a C18-bonded silica gel column (250 mm length, 46 mm inner diameter, five µm particle size). The mobile phase consisted of water (A) and acetonitrile (B) with a gradient elution as follows: 0–5 min, 38%–55% acetonitrile; 5–15 min, 55%–70% acetonitrile; 15–17 min, 38%–55% acetonitrile; 17–20 min, 70%–38% acetonitrile. The flow rate was 1.0 mL/min, the column temperature was set at 25 °C, the drift tube temperature at 105 °C, and the gas flow rate at 3.0 L/min. The retention time for Ophiopogonin D was 12.3 min.

For the extraction of methylophiopogonanone A and methylophiopogonanone B, 3.0 g of Ophiopogon japonicus root tuber powder was placed in a conical flask with 25 mL of water and allowed to stand for 24 h. The mixture was then ultrasonicated (500 W, 40 kHz) for 60 min, shaken well, and filtered to obtain the filtrate (Wu et al., 2016).

The chromatographic conditions for methylophiopogonanone A and methylophiopogonanone B also utilized a C18-bonded silica gel column (250 mm length, 46 mm inner diameter, five µm particle size). The mobile phase consisted of water (A) and acetonitrile (B) at 60% acetonitrile, employing isoelution. Detection was carried out at 296 nm, with a flow rate of 1.0 mL/min and a column temperature of 25 °C. The retention times for methylophiopogonanone A and methylophiopogonanone B were 13.5 min and 15.3 min, respectively.

Statistical analysis

All data was processed using Excel. Stacked bar charts and radar charts were created using Origin 2024b. Additionally, principal component analysis (PCA) of the quality of Ophiopogon japonicus was conducted using Origin 2024b, producing a 3D PCA plot. Prior to calculating scores for different treatments, the quality data was standardized. Bar charts and stacked bar charts were created using GraphPad Prism 8.

Data standardization for growth indicators of corn and Ophiopogon japonicus was performed using the ‘Stats’ package in R. Clustering was conducted with the ‘Pheatmap’ package, which generated a circular clustering heatmap using complete linkage and Euclidean distance. A 2D PCA analysis was created using stats, while inter-group correlation analysis employed the Pearson method, utilizing the ‘psych’ package. Correlation plots were visualized using the ‘ggplot2’ package, along with additional visualizations created through the online platform http://www.chiplot.online (accessed September 27, 2024).

One-way and multi-factor analysis of variance (ANOVA) were performed using SPSS 26.0, with multiple comparisons conducted using Duncan’s new multiple range test.

Results

Effects of different top-dressing conditions on corn growth and yield

Effects of different nutrient management protocols on agronomic traits of corn are presented in Fig. 3. Under different nutrient management protocols, the height of corn reached between 288.76 and 306.58 cm per plant. Treatments T2 and T4 were significantly taller than T5, with increases of 6.17% and 4.91%, respectively. The type of nitrogen fertilizer had a significant effect on corn height, with corn treated with ammonium nitrate showing superior growth. The stem circumference varied between 7.96 and 9.20 cm per plant across the treatments, with treatments T1 and T4 significantly outperforming T2 by 15.64% and 14.25%, respectively. All three factors demonstrated a significant interactive effect on corn stem circumference. The number of leaves per plant ranged from 12.67 to 13.56, with T4 significantly exceeding T8 by 7.02%. Micronutrient fertilizers and topdressing showed a significant interactive effect on the number of corn leaves.

Figure 3 (A–C) Effects of different nutrient management protocols on agronomic traits of corn.

N, nitrogen fertilizers; M, micronutrient fertilizers; T, topdressing; F, F-statistic in multifactor analysis. Different lowercase letters indicate significant differences at the P < 0.05 level.

Effects of different nutrient management protocols on the accumulation of fresh biomass in corn are presented in Fig. 4. Table 2 shows the results of multivariate analysis of corn fresh biomass based on different nutrient management protocols. Under different nutrient management protocols, the fresh biomass of corn leaves reached between 28.13 and 39.38 t/hm2. Treatment T5 was significantly higher than the other treatments, surpassing them by 34.62% (T1), 40.00% (T2), 16.67% (T3), 25.00% (T4), 29.63% (T6), 12.90% (T7), and 12.90% (T8). Both the type of nitrogen fertilizer and the topdressing had significant effects on the fresh biomass of corn leaves, with a notable interaction among the three factors (N×M×T). The fresh biomass of corn stems ranged from 31.50 to 37.13 t/hm2, with treatments T4, T5, and T8 significantly higher than T3 by 17.86%, 17.86%, and 14.29%, respectively. There was a significant interaction between nitrogen and micronutrient treatments (N×M) regarding the fresh biomass of corn stems. For corn with husks, the fresh biomass ranged from 22.50 to 27.00 t/hm2, with treatments T3, T5, and T8 significantly exceeding T1 and T2. Both nitrogen and topdressing had significant effects on the fresh biomass of corn with husks, and these two factors exhibited a notable interaction.

Figure 4 Effects of different nutrient management protocols on the accumulation of fresh biomass in corn.

Different lowercase letters indicate significant differences at the P < 0.05 level. Different uppercase letters indicate significant differences in the total yield of corn (sum of different parts) at the P < 0.05 level.

Table 2 Multifactor analysis of fresh biomass of corn based on different nutrient management protocols.

Symbol	Fresh biomass of corn	Corn yield	
	Corn with husks	Corn leaf	Corn stem		
N	8.95**	9.96**	Ns	5.28*	
M	Ns	Ns	Ns	Ns	
T	17.54***	18.53***	Ns	18.20***	
N×M	Ns	Ns	12.42**	8.72**	
N×T	8.95**	Ns	Ns	Ns	
M×T	Ns	Ns	Ns	Ns	
N×M×T	Ns	6.67*	Ns	Ns	
Notes.

N means types of nitrogen fertilizers, M means micronutrient fertilizers, and T means topdressing.

In the multifactor analysis, ∗ represents P < 0.05, ∗∗ indicates P < 0.01, and ∗∗∗ signifies P < 0.001. Ns, not significant.

Overall, the fresh yield of silage corn varied from 84.38 to 103.50 t/hm2, with T5 significantly outperforming the other treatments (except for T7 and T8). T5 exceeded the other treatments by 16.46% (T1), 22.67% (T2), 12.20% (T3), 12.20% (T4), and 15.00% (T6). Both nitrogen and topdressing had significant effects on the fresh yield of silage corn, with a significant interaction between nitrogen and micronutrient treatments (N×M).

The circular clustering diagram illustrates the effects of different nutrient management protocols on agronomic traits and fresh biomass accumulation in corn is presented in Fig. 5. According to the circular clustering diagram of corn, the variables corn plant height, corn stem girth, and number of corn leaves formed one cluster, while the fresh biomass of corn and corn yield from different parts grouped together in another cluster. The results of the hierarchical clustering analysis indicate that the eight treatments can be divided into three categories. The first category consists solely of T5, where the fresh biomass of corn parts and corn silage yield performed the best, while corn plant height, stem girth, and number of leaves showed the worst results. The second category includes T7 and T8, where the fresh biomass and corn silage yield were inferior compared to the first category, but corn plant height and stem girth exhibited improvements relative to the first cluster. The remaining treatments, T1, T2, T3, T4, and T6, fall into the third category, which overall showed relatively high values for corn plant height, stem girth, and number of leaves, but the lowest performance for the fresh biomass of corn parts and corn silage yield.

Figure 5 Circular clustering diagram of corn growth indices of different nutrient management protocols.

The PCA diagram illustrates the effects of different nutrient management protocols on corn agronomic traits and fresh biomass accumulation is shown in Fig. 6. PCA analysis was conducted using corn agronomic traits, biomass from different parts, and corn silage yield. The first principal component (PC1) accounted for 45.62% of the variance, while the second principal component (PC2) accounted for 18.62%, resulting in a cumulative variance contribution of 64.24% for the two principal components. The P-value of 0.2230 indicates that the differences between groups are not significant.

Figure 6 Principal component analysis (PCA) plot of corn.

In the analysis, T5 was categorized as a distinct group, while T7 and T8 were grouped together. The remaining treatments, T1, T2, T3, T4, and T6, formed another cluster. These findings are consistent with the results of the circular clustering analysis.

Effects of different nutrient management protocols on the growth and quality of Ophiopogon japonicus

Effects of different nutrient management protocols on the growth of Ophiopogon japonicus

Effects of different nutrient management protocols on agronomic traits of Ophiopogon japonicus are presented in Fig. 7. Under different top-dressing treatments, the number of root tubers for Ophiopogon japonicus ranged from 12.7 to 16.4, with T8 significantly higher than T1 by 29.65%. The micronutrient fertilizers (M) had a significant impact on the number of root tubers. The plant height of Ophiopogon japonicus reached 22.9 to 26.7 cm/plant, with T1 and T4 significantly taller than T5, T7, and T8. The number of nutritive roots varied from 6.6 to 9.8, where T2 and T4 were significantly higher than T7 by 38.24% and 48.45%, respectively, with no significant differences among the other treatments. The topdressing (T) significantly influenced both plant height and the number of nutritive roots. The storage root count ranged from 16.8 to 19.8, with no significant differences observed between treatments.

Figure 7 Effects of different nutrient management protocols on agronomic traits of Ophiopogon japonicus (OJ).

Effects of different nutrient management protocols on the accumulation of fresh biomass indifferent parts of Ophiopogon japonicus are presented in Fig. 8. Table 3 shows the results of multivariate analysis of Ophiopogon japonicus fresh biomass based on different nutrient management protocols. Under different nutrient management protocols, the fresh biomass of root tubers ranged from 11.00 to 17.91 t/hm2, with T4 to T8 significantly higher than T1 to T3, achieving increases of 48.86% to 62.75% compared to T1, and 18.45% to 29.50% compared to T2 and T3. The types of nitrogen fertilizers, micronutrient fertilizers, and topdressing all had significant effects on the fresh biomass of root tubers, with notable interactions between N and T.

Figure 8 Effects of different nutrient management protocols on the accumulation of fresh biomass in different parts of Ophiopogon japonicus.

Different lowercase letters indicate significant differences at the P < 0.05 level. Different uppercase letters indicate significant differences in the total yield of corn (sum of different parts) at the P < 0.05 level.

The fresh biomass of Ophiopogon japonicus leaves ranged from 19.50 to 23.45 t/hm2, while the fresh biomass of rhizomes varied from 2.22 to 3.21 t/hm2. For nutritive roots, it ranged from 2.88 to 4.69 t/hm2, and storage roots ranged from 6.63 to 9.27 t/hm2. Among these, the N, M, and T showed significant interaction effects only on the fresh biomass of storage roots, with no significant impact on the other indicators.

The circular clustering diagram illustrates the effects of different nutrient management protocols on agronomic traits and fresh biomass accumulation in Ophiopogon japonicus is presented in Fig. 9. According to the circular clustering analysis of Ophiopogon japonicus, the fresh biomass of root tubers, OJ storage roots, and OJ root tubers were grouped together, while OJ plant height and fresh biomass of OJ leaves formed another group. A third group included OJ nutritive roots, fresh biomass of OJ nutritive roots, fresh biomass of OJ rhizomes, and fresh biomass of OJ storage roots.

The results indicate that the eight treatments can be categorized into three main groups. The first group consists of T8, which performed optimally in terms of root quantity and biomass but showed poorer results in OJ plant height and fresh biomass of OJ leaves. The second group includes T5, T6, and T7, which excelled primarily in fresh biomass of root tubers but were less effective in other measures. The remaining treatments, T1, T2, T3, and T4, form the third group, which exhibited overall poorer performance in fresh biomass of root tubers, OJ storage roots, and OJ root tubers, while showing better results in OJ plant height, fresh biomass of OJ leaves, OJ nutritive roots, fresh biomass of OJ nutritive roots, fresh biomass of OJ rhizomes, and fresh biomass of OJ storage roots.

The PCA diagram illustrating the effects of different nutrient management protocols on Ophiopogon japonicus agronomic traits and fresh biomass accumulation is shown in Fig. 10. In the PCA analysis based on Ophiopogon japonicus plant height, root quantity, and biomass of various parts, PC1 accounted for 34.91% and PC2 accounted for 24.10%, resulting in a cumulative variance contribution rate of 59.01%. The p-value was 0.2670, indicating no significant differences between the groups. Treatment T8 was categorized as a separate group, while treatments T5, T6, and T7 formed another group, and treatments T1, T2, T3, and T4 were grouped together. This classification aligns with the results of the circular clustering analysis.

Table 3 Multifactor analysis of fresh biomass of Ophiopogon japonicus based on different nutrient management protocols.

Symbol	Fresh biomass of Ophiopogon japonicus	
	Leaf	Rhizome	Storage root	Nutritive root	Root tuber	
N	Ns	Ns	Ns	Ns	9.72**	
M	Ns	Ns	Ns	Ns	13.47**	
T	Ns	Ns	Ns	Ns	43.12***	
N×M	Ns	Ns	Ns	Ns	Ns	
N×T	Ns	Ns	Ns	Ns	5.55*	
M×T	Ns	Ns	Ns	Ns	Ns	
N×M×T	Ns	Ns	5.26*	Ns	Ns	
Notes.

N means types of nitrogen fertilizers, M means micronutrient fertilizers, and T means topdressing. In the multifactor analysis, * represents P < 0.05, ** indicates P < 0.01, and *** signifies P < 0.001. Ns, not significant.

Figure 9 Circular clustering diagram of Ophiopogon japonicus (O]) growth indices of different nutrient management protocols.

Figure 10 PCA plot of Ophiopogon japonicus growth indices.

Effects of different top-dressing conditions on the quality of Ophiopogon japonicus

Effects of different nutrient management protocols on the quality of Ophiopogon japonicus are presented in Fig. 11. Table 4 shows the results of multivariate analysis of quality indices for Ophiopogon japonicus based on different nutrient management protocols. Under different nutrient management protocols, the extract reached 82.78% to 85.03%, with T2 significantly higher than other treatments (except T3) by 1.31% to 2.72%. Nitrogen type (N) and topdressing (T) had significant effects on the extract, and there was a significant interaction between N and micronutrient fertilizers (M). Total polysaccharide content ranged from 61.52% to 67.54%, with T1 and T3 significantly higher than T2, T7, and T8 by 4.37% to 9.79%. Micronutrient fertilizers (M) and topdressing (T) had significant effects on total polysaccharides, and there were significant interactions among N, M, and T. Total saponin content varied from 0.15% to 0.23%, with T4, T7, and T8 significantly higher than other treatments by 30.13% to 56.72%. All factors (N, M, T) significantly affected total saponin, with significant interactions among all factors except for the interaction between M and T. Total flavonoid content ranged from 0.16% to 0.19%, with T7 significantly higher than other treatments by 5.89% (T1), 3.93% (T2), 22.12% (T3), 14.60% (T4), 3.47% (T5), 8.24% (T6), and 9.70% (T8). N, M, and T significantly influenced total flavonoid content, with significant interactions among all factors except for the interaction between N and M. Ophiopogonin D levels reached 0.19 to 0.48 mg/g, with T4 and T7 significantly higher than other treatments, where T4 was 36.59% to 150.42% higher and T7 was 27.52% to 133.79% higher than others. All factors (N, M, T) significantly affected Ophiopogonin D, and significant interactions were observed among all factors. methylophiopogonanone A ranged from 52.69 to 103.88 µg/g, with T5 significantly higher than other treatments (except T7) by 63.29% (T1), 21.48% (T2), 97.17% (T3), 38.86% (T4), 17.69% (T6), and 50.25% (T8). Micronutrient fertilizers (M) and topdressing (T) significantly affected methylophiopogonanone A, with a significant interaction between N and T. methylophiopogonanone B varied from 32.37 to 65.51 µg/g, with T6 significantly higher than other treatments (except T5) by 56.55% (T1), 29.29% (T2), 102.39% (T3), 44.79% (T4), 14.23% (T7), and 59.27% (T8). Both M and T significantly influenced methylophiopogonanone B, with significant interactions among all factors except for the interaction between N and M.

Figure 11 Effects of different nutrient management protocols on quality of Ophiopogon japonicus.

Table 4 Multivariate analysis of quality indices for Ophiopogon japonicus based on different nutrient management protocols.

Symbol	Extract	Total polysaccharide	Total saponin	Total flavonoid	Ophiopogon saponin D	MOPA	MOPB	
N	6.61*	9.86**	15.19**	13.32**	6.44*	Ns	Ns	
M	Ns	Ns	91.39***	133.92***	189.19***	25.50***	42.38***	
T	34.64***	13.59**	14.42**	148.79***	36.68***	68.09***	67.55***	
N×M	14.63**	Ns	16.51**	Ns	7.22*	Ns	Ns	
N×T	Ns	Ns	12.70**	143.75***	69.38***	78.36***	26.49***	
M×T	Ns	8.11*	Ns	204.01***	15.86**	Ns	5.02*	
N×M×T	Ns	6.11*	10.66**	24.01***	42.52***	Ns	10.17**	
Notes.

N means types of nitrogen fertilizers, M means micronutrient fertilizers, and T means topdressing. In the multifactor analysis, * represents P < 0.05, ** indicates P < 0.01, and *** signifies P < 0.001. MOPA means Methylophiopogonanone A, MOPB means Methylophiopogonanone B. Ns, not significant.

The circular clustering diagram demonstrating the effects of different nutrient management protocols on quality indices of Ophiopogon japonicus is presented in Fig. 12. Based on the circular clustering heatmap of Ophiopogon, Extract and Total polysaccharide were grouped together, while Total saponin and Ophiopogonin D formed another group. Total flavonoid, methylophiopogonanone A, and methylophiopogonanone B were clustered as a third group. The hierarchical clustering analysis indicated that the eight treatments could be broadly categorized into three groups. The first group included T4 and T8, which exhibited the best overall performance in total saponin and Ophiopogonin D but performed poorly in total flavonoid, methylophiopogonanone A, and methylophiopogonanone B. The second group comprised T5, T6, and T7, which showed superior results in total flavonoid, methylophiopogonanone A, and methylophiopogonanone B, but had lower performance in other aspects. The remaining treatments formed the third group, including T1, T2, and T3, which performed well in Extract and Total polysaccharide but were less effective in the other five quality indicators.

Figure 12 Circular clustering diagram of Ophiopogon japonicus quality indices.

Correlation analysis of the effects of different nutrient management protocols on agronomic traits and fresh biomass accumulation in Ophiopogon japonicus is presented in Fig. 13. In the correlation analysis, Total saponin and Ophiopogonin D showed a highly significant positive correlation with Ophiopogon japonicus (OJ) yield, while methylophiopogonanone A demonstrated a significant positive correlation with OJ yield. Extract was significantly positively correlated with both OJ nutritive root and fresh biomass of OJ rhizome. Conversely, Extract and Total polysaccharide exhibited a significant negative correlation with OJ yield. Additionally, Total flavonoid and methylophiopogonanone A showed a significant negative correlation with OJ plant height, and Total flavonoid was significantly negatively correlated with OJ nutritive root.

Figure 13 Correlation analysis of growth parameters in Ophiopogon japonicus under different nutrient management protocols.

Effects of different nutrient management protocols on the grading of Ophiopogon japonicus are presented in Fig. 14. Under different nutrient management protocols, the first-grade Ophiopogon japonicus yield reached 33.85% to 40.77%, with T3 significantly outperforming T1, T2, T4, and T5 by 7.64% (T1), 8.39% (T2), 20.43% (T4), and 13.56% (T5), while showing no significant correlation with T6, T7, and T8. The second-grade yield varied between 47.39% and 54.08%, with T4 and T5 significantly higher than the other treatments, where T4 exceeded other groups by 4.27% to 12.42%, and T5 by 5.85% to 14.13%. The third-grade yield ranged from 10.02% to 14.55%, with T1 significantly higher than all other groups by 28.83% (T2), 35.75% (T3), 13.04% (T4), 45.28% (T5), 31.57% (T6), 32.30% (T7), and 15.26% (T8).

Figure 14 Effects of different nutrient management protocols on the grading of Ophiopogon japonicus.

The 3D PCA plot of quality indicators and medicinal properties of Ophiopogon japonicus is shown in Fig. 15. Table 5 shows the results of PCA of the quality indicators and medicinal properties of Ophiopogon japonicus. PCA was performed on the quality indicators and medicinal properties of Ophiopogon japonicus root tuber.

Figure 15 3D PCA of the quality indicators and medicinal properties of Ophiopogon japonicus.

MOPA means methylophiopogonanone A, MOPB means methylophiopogonanone B.

Table 5 PCA of the quality indicators and medicinal properties of Ophiopogon japonicus.

Traits	PC 1	PC 2	PC 3	
Eigenvalue	3.290	2.067	0.821	
Variability (%)	47.00	29.53	11.73	
Cumulative (%)	47.00	76.53	88.26	
Extract	Factor loadings	−2.781	0.539	−0.257	
Total polysaccharide	−2.047	0.522	1.724	
Total saponin	1.775	−2.147	−0.001	
Total flavonoid	2.710	1.274	−0.345	
Ophiopogonin D	1.956	−1.901	0.667	
Flavanone A	3.207	0.983	0.293	
Flavanone B	2.835	1.232	0.532	
Extract	Component Score Coefficient Matrix (CSC)	−0.417	0.150	−0.129	
Total polysaccharide	−0.307	0.145	0.865	
Total saponin	0.266	−0.598	−0.001	
Total flavonoid	0.406	0.355	−0.173	
Ophiopogonin D	0.293	−0.529	0.335	
Flavanone A	0.480	0.274	0.147	
Flavanone B	0.425	0.343	0.267	

The first principal component (PC1) accounted for 47.00%, the second (PC2) for 29.53%, and the third (PC3) for 11.73% of the variance. Together, these three principal components explained 88.26% of the variance, representing most of the information content in the quality and medicinal indicators of Ophiopogon japonicus root tuber. Therefore, these three principal components can be used to evaluate the quality and medicinal properties of Ophiopogon japonicus root tuber from the eight treatment groups.

PC1 primarily integrates the information from extract, total polysaccharide, total saponin, total flavonoid, Ophiopogonin D, methylophiopogonanone A and methylophiopogonanone B. These indicators show a positive distribution on the first principal component, meaning that the higher the PC1 value, the higher the values of these indicators. PC2 mainly reflects the total saponin.

Based on the PCA model, the eigenvectors for each indicator were calculated and used as weights to derive the scoring formulas for the three principal components (H1, H2, H3). SPSS software was used to standardize the quality data of the Ophiopogon japonicus root tuber and, in putting it into these scoring formulas, we obtained the scores of Ophiopogon japonicus root tuber from different treatment groups on seven principal components. The weights for these principal components were based on their respective variance contribution rates, and the comprehensive quality score (H0) of the Ophiopogon japonicus root tuber was calculated accordingly.

H1 = −0.417Z1 − 0.307Z2 + 0.266Z3 + 0.406Z4 + 0.293Z5 + 0.480Z6 + 0.425Z7

H2 = 0.150Z1 + 0.145Z2 − 0.598Z3 + 0.355Z4 − 0.529Z5+ 0.274Z6 + 0.343Z7

H3 = −0.129Z1 + 0.865Z2 − 0.001Z3 − 0.173Z4 + 0.335Z5 + 0.147Z6 + 0.267Z7

H0 = 0.533H1 + 0.335H2 + 0.133H3

The comprehensive scores for each treatment group are ranked as follows: T5 > T7 > T6 > T2 > T1 > T8 > T4 > T3 (Table 6). Among the eight treatment groups, the Ophiopogon japonicus root tuber from T5 treatment group has the highest quality. The Ophiopogon japonicus root tuber from the treatment groups of T1, T8, T4, and T3 have relatively poor quality.

Table 6 Score table of the PCA for the quality indicators and medicinal properties of Ophiopogon japonicus.

Varieties	H 1	H 2	H 3	H 0	
T1	−1.555	1.047	0.203	−0.451	
T2	−0.598	1.664	−1.083	0.095	
T3	−3.207	−0.637	0.391	−1.871	
T4	−0.013	−2.147	0.918	−0.604	
T5	1.899	1.142	0.463	1.456	
T6	0.472	1.092	0.476	0.680	
T7	2.406	−0.465	0.356	1.174	
T8	0.596	−1.696	−1.724	−0.480	

Discussion

Effective ecological planting models in modern agriculture integrate various components to achieve sustainability, economic viability, and environmental protection. These models emphasize the harmonious interaction between agricultural activities and natural ecosystems, offering significant advantages in resource utilization, soil improvement, pest and weed suppression, and enhanced economic and ecological benefits. However, they also present challenges such as complex management and resource competition. Despite these challenges, ecological planting models hold great potential for contributing to sustainable development and environmental conservation, making them a promising approach for the future cultivation of medicinal plants.

As the primary approach to cultivating medicinal plants, ecological planting is increasingly recognized for its role in enhancing the sustainability of traditional Chinese medicine (TCM) production and improving environmental management. This method integrates ecological principles with agricultural practices to ensure high-quality medicinal materials while minimizing environmental impact. Currently, ecological planting models for medicinal plants include simulated wild cultivation (Schafer, 2011), understory planting (He et al., 2022b), intercropping (Kang et al., 2020b), crop rotation, and biodiversity-based landscape farming (Kang et al., 2020a). Among these, intercropping is the most widely adopted production method. By optimizing spatial and temporal arrangements, intercropping efficiently utilizes natural resources such as land, water, nutrients, sunlight, and labor (Latati et al., 2013). This approach not only maximizes land use efficiency and increases crop yields but also minimizes ecological disturbances. To fully realize the benefits of intercropping, scientific planning, strategic crop selection, and enhanced resource and nutrient management are essential. Proper implementation can maximize productivity while mitigating potential drawbacks, ensuring a balanced and sustainable agricultural system.

Fertilizers are essential for crop growth and play a critical role in ensuring food security. Nitrogen (N), phosphorus (P), and potassium (K) are the primary macronutrients required for plant development, making fertilizers a key input in agricultural production. However, in intercropping systems, nutrient competition between crops can limit the efficient utilization of available nutrients (Nasar et al., 2021; Nyawade et al., 2021). In the Ophiopogon japonicus-corn intercropping system, Ophiopogon japonicus primarily undergoes its seedling stage during intercropping. At this stage, the crop focuses on leaf and fibrous root growth, with nutrient accumulation mainly in these organs. Meanwhile, corn is in a vigorous growth phase, resulting in high nutrient demands, particularly for nitrogen. Studies indicate that both corn and Ophiopogon japonicus require substantial nitrogen inputs, with corn, as a grass species, having a competitive advantage in nitrogen uptake. Intercropping further influences the nitrogen-use efficiency of both crops (Li et al., 2001). Nitrogen plays a crucial role in chlorophyll synthesis, enzyme activity, and overall plant metabolism (Pan et al., 2022). Proper nitrogen supplementation can mitigate the negative effects of nutrient competition in intercropping systems. During the co-growth period, corn’s nitrogen demand peaks during its rapid growth and grain formation stages, while Ophiopogon japonicus requires less nitrogen at the seedling stage. Moderate nitrogen application supports its leaf growth and root development, but excessive nitrogen can lead to excessive vegetative growth, reducing root yield and quality. To balance the nitrogen demands of both crops, fast-acting nitrogen fertilizers are preferred. Urea and ammonium nitrate are two common nitrogen sources. Urea provides amide nitrogen, which must be converted into ammonium or nitrate by soil microbes before plant uptake. In contrast, ammonium nitrate supplies both ammonium and nitrate nitrogen, which are readily available for plant absorption. As a result, ammonium nitrate generally has a higher nitrogen-use efficiency than urea (Sylvester-Bradley et al., 2014). Our study found that nitrogen supplementation significantly improved the growth of both crops, increasing corn silage and Ophiopogon japonicus root yields while maintaining root quality. These findings align with previous research on intercropping systems, such as corn-alfalfa and corn-soybean intercropping (Nasar et al., 2020; Shao et al., 2020). Notably, nitrogen fertilizer type had a significant effect on corn plant height, with ammonium nitrate yielding better results. This is attributed to its fast-acting nitrogen availability, whereas urea requires microbial conversion before plant uptake (Song et al., 2023; Waterhouse et al., 2017). Prior studies have demonstrated the impact of nitrogen forms on crop growth and yield, particularly in corn (Reyes-Matamoros et al., 2024) and tobacco (Li et al., 2024b).

Nitrogen (N), phosphorus (P), and potassium (K) are essential nutrients for plant growth, directly influencing crop yield and quality. However, different intercropping systems have varying nutrient demands and absorption patterns (Nasar et al., 2020; Zaeem et al., 2019). The optimal NPK ratio is critical for maximizing plant growth and yield. Excessive or unbalanced use of chemical fertilizers can lead to nutrient imbalances, reduced efficiency, soil salinization, and water pollution (He et al., 2022a). Only through the interaction of N, P, and K can root development, biomass production, and overall plant health be effectively regulated. The NPK ratio serves as a key indicator of nutrient limitations. A deficiency in phosphorus or potassium can reduce nitrogen efficiency, ultimately inhibiting plant growth and yield (Usherwood & Segars, 2001). For instance, root and tuber crops require higher potassium levels to support tuber development and water regulation, while a balanced supply of nitrogen and phosphorus is crucial for strong root growth and nutrient uptake (Smith, 1976). The impact of different NPK ratios on medicinal plant cultivation is multifaceted, affecting both growth and medicinal quality. Studies have shown that varying NPK ratios influence biomass accumulation in licorice (Chen et al., 2023), with higher nitrogen, phosphorus, and potassium levels promoting dandelion biomass while lower levels favor secondary metabolite accumulation (Su et al., 1994). Additionally, specific NPK ratios significantly impact the accumulation of phenolic compounds in Rhodiola rosea, with an optimized ratio enhancing salidroside and other phenolic contents (Zhao et al., 2018). A 3:1:3 NPK combination has been shown to increase ginsenoside content and improve rhizosphere microbial diversity, benefiting plant health (Sun et al., 2022). Optimizing nutrient balance can significantly enhance both biomass production and active compound concentration, which is vital for medicinal efficacy. Different NPK ratios also play a crucial role in intercropping systems, influencing crop growth, yield, and quality traits. Applying specific NPK ratios can improve productivity and resource use efficiency. For example, in a rice-soybean intercropping system, a higher NPK ratio significantly enhanced plant height, leaf area index, and seed weight, leading to increased rice yield components (Effam et al., 2024). Similarly, intercropping cotton with peanut, garlic, or wheat increased aboveground biomass by 15.5%, with improved nutrient uptake contributing to yield advantages (Qiu et al., 2023). In foxtail millet, the optimal NPK ratio (N160P90K150) significantly boosted yield and quality traits, highlighting the importance of balanced fertilization in intercropping systems (Xing et al., 2023). In this study, we examined the effects of different NPK top-dressing ratios on the growth, yield, and quality traits of an Ophiopogon japonicus-corn intercropping system. Corn has high nutrient demands, particularly for nitrogen (from the jointing to tasseling stage), phosphorus (during the seedling and jointing stages), and potassium (from the jointing to grain-filling stage). Meanwhile, Ophiopogon japonicus requires higher phosphorus and potassium levels during its seedling stage for root growth and tillering. Our results indicate that different NPK ratios significantly affect the growth, yield, and quality of both crops, aligning with previous research findings. While an optimal NPK ratio can enhance crop quality and yield, its effects vary depending on crop species and environmental conditions, suggesting a complex interaction between nutrient utilization and plant growth dynamics (Jose, 2022). Furthermore, soil conditions and crop-specific requirements influence fertilizer effectiveness, emphasizing the need for tailored fertilization strategies for different cropping systems (Singh et al., 2015).

Micronutrients play a crucial role in plant growth, acting as catalysts and participants in various physiological and biochemical processes. While they do not directly influence crop growth as macronutrients do, they are essential for numerous metabolic functions. Magnesium (Mg) is a vital micronutrient for corn growth, playing a key role in physiological processes. It is the core element of chlorophyll formation (El-Dissoky, Al-Kamar & Derar, 2017) and serves as a cofactor for several carbon fixation-related enzymes, influencing amino acid and lipid phosphorylation and synthesis (Cai et al., 2022). Additionally, magnesium is essential for protein synthesis (El-Dissoky, Al-Kamar & Derar, 2017). Additionally, magnesium is essential for protein synthesis (Maathuis, 2009). The nutritional status of magnesium fertilizers affects the transport of photosynthetic products from leaves to growing roots, influencing root nutrient absorption capacity (Chen et al., 2017; Kiss, Stefanovits-Bányai & Takács-Hájos, 2004). Notably, magnesium can reduce the concentration of other micronutrients in plants, not only due to a biomass dilution effect but also by altering root absorption capacity (Kleiber, Golcz & Krzesinski, 2012). Despite its importance, magnesium is often overlooked in fertilization programs compared to other nutrients. Recent studies have highlighted the benefits of magnesium fertilization, particularly in improving corn yield and nutrient uptake. For example, magnesium nanoparticles have been shown to enhance corn Mg absorption and utilization efficiency, increase phosphorus, potassium, and calcium content in harvested grains, and improve yield and phenological development (Zhang et al., 2020). Manganese (Mn) is another critical micronutrient for corn, playing a vital role in metabolism and growth. It is essential for physiological processes such as photosynthesis, respiration, and nitrogen assimilation. Manganese availability and concentration influence nutritional value and agricultural productivity. Specialized manganese fertilizers have been proven to enhance corn yield and stress resistance. For instance, high-nitrogen compound fertilizers enriched with manganese can address nutrient imbalances, promote crop growth, increase yield, and improve soil fertility (Osman, 2013). Our study demonstrates that magnesium and manganese fertilization significantly improved the yield of silage corn and Ophiopogon japonicus. Notably, magnesium and manganese application increased the number of Ophiopogon japonicus tuberous roots, leading to a higher yield of Ophiopogonis Radix. However, the increased yield resulted in a slight decline in quality (Fan et al., 2023). Therefore, in addition to nitrogen, phosphorus, and potassium, balanced fertilization strategies should include appropriate supplementation of essential micronutrients based on crop requirements. This approach not only enhances yield but also improves medicinal material quality. Inappropriate fertilization can hinder plant growth, reduce yield, and decrease the content of bioactive compounds. However, further research is needed to explore the photosynthetic efficiency of nitrogen, phosphorus, potassium, magnesium, and manganese in the corn-Ophiopogon japonicus intercropping system.

The quality of Ophiopogonis Radix is significantly influenced by the types of nitrogen fertilizers and topdressing applications, affecting key bioactive compounds such as extract yield, total polysaccharides, total saponins, and total flavonoids. Interactions between different factors also lead to variations in the medicinal components, including Ophiopogonin D, methylophiopogonanone A, and methylophiopogonanone B. These findings align with previous fertilization studies on monocropped Ophiopogon japonicus, which identified an optimal combination of organic and inorganic fertilizers to enhance bioactive compounds while minimizing heavy metal accumulation. This may be attributed to the application of nitrogen and potassium fertilizers, which enhances carbon assimilation enzyme activity, promotes carbohydrate synthesis, and facilitates the formation of terpenoid compounds that rely on non-structural carbohydrates as precursors (Waring, Perkowski & Smith, 2023). A similar pattern has been observed in Panax notoginseng, where high nitrogen levels suppress the accumulation of saponins-a key bioactive compound-by reducing nitrogen use efficiency and photosynthetic capacity. Conversely, low nitrogen conditions promote the accumulation of saponins and flavonoids by increasing the C/N ratio and upregulating genes involved in their biosynthesis (Cun et al., 2023). This highlights the importance of optimizing nitrogen levels to balance plant growth with the production of high-quality medicinal metabolites.

In response to the challenges faced by Ophiopogonis Radix production, the adoption of an Ophiopogon japonicus-corn intercropping system presents an effective solution. This approach not only improves land use efficiency but also mitigates the competition between medicinal and food crops caused by large-scale Ophiopogon japonicus cultivation. However, nutrient competition between Ophiopogon japonicus and corn necessitates precise nutrient supplementation during cultivation. To ensure both yield and quality, it is crucial to regulate the supply and form of macronutrients such as nitrogen, phosphorus, and potassium while also addressing the availability of essential micronutrients like magnesium and manganese. These elements play a vital role in plant growth, stress resistance, and overall crop performance, further enhancing the benefits of intercropping. By implementing a well-managed nutrient strategy, this model can achieve both medicinal and food crop production while supporting sustainable agriculture, offering valuable insights for similar cropping systems.

Conclusions

This study explores the potential of the corn-Ophiopogon japonicus intercropping system in improving land use efficiency and alleviating competition between medicinal and food crops. Additionally, it proposes an optimized fertilization strategy to enhance both crop yield and quality. The results indicate that appropriate nutrient supplementation not only increases the yield of silage corn and Ophiopogon japonicus but also improves the quality of Ophiopogonis Radix. For yield improvement, the recommended fertilization plan includes ammonium nitrate (150 kg N/ha), magnesium sulfate (45 kg/ha), manganese sulfate (15 kg/ha), superphosphate (75 kg P2O5/ha), and potassium sulfate (450 kg K2O/ha). This combination enhances corn agronomic traits such as plant height and stem diameter, significantly increasing silage corn yield while also promoting Ophiopogon japonicus tuber formation, thereby boosting Ophiopogonis Radix production. To improve the medicinal quality of Ophiopogonis Radix, the study recommends applying urea (150 kg N/ha), superphosphate (75 kg P2O5/ha), and potassium sulfate (450 kg K2O/ha), which ensures stable yield while significantly enhancing the bioactive compound content.

In summary, this study provides scientific evidence for nutrient management in the corn-Ophiopogon japonicus intercropping system, demonstrating that optimized fertilization strategies can achieve a win-win outcome for both food and medicinal crops. Future research should explore: (1) the adaptability of this intercropping system across different soil types and climatic conditions; (2) the long-term impact of the recommended fertilization strategy on soil health and ecosystem stability; (3) the integration of modern precision agriculture technologies, such as remote sensing and intelligent fertilization systems, to further refine nutrient management; and (4) the mechanisms of crop interactions in intercropping, particularly the role of root exudates and microbial communities in nutrient uptake and plant growth. These studies will provide a more comprehensive theoretical foundation and technical support for the wider adoption of the corn–Ophiopogon japonicus intercropping system.

Additionally, future research should assess the economic and social benefits of this system, including its potential to reduce fertilizer use, lower environmental pollution, and increase farmers’ income. Such insights could contribute to sustainable agricultural development by offering innovative solutions and practical strategies.

Supplemental Information

Supplemental Information 1 Raw data

Special thanks to Ying Liu from Sichuan Zhijiacheng Biotechnology Co., Ltd. for her support and assistance during the field trial of this project.

Additional Information and Declarations

Competing Interests

Author Contributions

Data Availability

The authors declare there are no competing interests.

Xiaoyang Cai conceived and designed the experiments, analyzed the data, prepared figures and/or tables, authored or reviewed drafts of the article, and approved the final draft.

Heling Fan conceived and designed the experiments, analyzed the data, prepared figures and/or tables, authored or reviewed drafts of the article, and approved the final draft.

Hongmei Deng performed the experiments, authored or reviewed drafts of the article, and approved the final draft.

Wenjing Li performed the experiments, authored or reviewed drafts of the article, and approved the final draft.

Haohan Wang performed the experiments, analyzed the data, prepared figures and/or tables, authored or reviewed drafts of the article, and approved the final draft.

Jiaming Zhang performed the experiments, authored or reviewed drafts of the article, and approved the final draft.

Min Li conceived and designed the experiments, authored or reviewed drafts of the article, and approved the final draft.

The following information was supplied regarding data availability:

The raw measurements are available in the Supplementary File.

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
