# Peer review of "Optimizing nutrient management protocol for Ophiopogon japonicus-corn intercropping: impacts on growth, yield, and medicinal quality"

_PeerJ, doi:10.7717/peerj.19655_

## Round 0.1 · original submission · Major Revisions

Kindly revise your manuscript according to the reviewers' comments.
Extend your discussion as more details are required to correlate the results of different studies.
Revise the grammar of your manuscript.

Reviewer 1 ·

Basic reporting

No comment.

Experimental design

I have indicated in the text the applications that need to be defined in the Method/experimental design section. It should be added.

Validity of the findings

The summary scheme in bullet points at the end of the discussion section should be moved to the conclusion section. I stated it in the text.

Annotated reviews are not available for download in order to protect the identity of reviewers who chose to remain anonymous.

·

Basic reporting

- check species' scientific names throughout the manuscript
- references are suitables
- Article well structured
Please see additional comments section

Experimental design

Experimental design is well structured
Methods are well- described
Please see additional comments section

Validity of the findings

Some details must be added
Please see additional comments section

Additional comments

Manuscript ID: #110675

Manuscript title: The effects of different topdressing conditions on the growth, yield, and medicinal quality of Ophiopogon japonicus and corn in an intercropping system
General comments
The article is generally well-structured and well-developed, with some remarks to consider. I congratulate the authors on the quality of the manuscript.
Specific comments
Title
- I think the title is suitable. However, it would be better if it focused on what you found that is novel or important, not what you did.
Abstract
- An introduction sentence underlying the problematic must be added at the beginning
Keywords
- The words used in the title must not be repeated in keywords.
Introduction
- Please add a section about the knowledge gap related to adopting intercropping to ameliorate crop growth, yield, and medicinal quality.
Material & methods
- Sometimes, ‘Ophiopogon japonicus’ is not italicized in the text (Line 145). Please check it
- Figures: Self-representation is required in figures and tables. Please ensure that all abbreviations are explained within the figures.
Results
- The PCA biplots adopted describe only relatioships between treatments, it is better to adopt a PCA biplots including treatments and measured traits.
Conclusion
- "Conclusion" section is the final section of your research paper. This is the perfect place to convey the final take-home message to your readers. In this section, the authors answer the initial scientific question, indicate the relevant results, and propose future studies (perspectives) to complete their research. Please rephrase this section according to these suggestions.

·

Basic reporting

Comments: Your manuscript entitled: “The effects of different topdressing conditions on the growth, yield, and medicinal quality of Ophiopogon japonicus and corn in an intercropping system” has been carefully reviewed. It is an interesting study but needs major revisions before acceptance. Here are the most important comments on the text and in the manuscript:

The entire manuscript is long. Rephrase the manuscript. Do not use T1, T2, ... because it makes it difficult to understand. Since the study involved three factors, it should be written in a way that makes it easier to understand.
Correct hyphen misuse throughout manuscript.

The title does not reflect the content. I suggest referring to fertilization practices and intercropping. The use of the term topdressing made the manuscript confusing.

The abstract mainly contains a description of the results, it does not provide greater insight into the results.

Keywords Some of the keywords are already present in the title and others are very general. It would be better if authors consider different keywords.

Introduction:
Your introduction needs more detail about intercropping. The text is largely based on the medicinal use of O. japonicus. I suggest that the authors initially give greater depth to intercropping system and fertilization of N, P and micronutrients.

Methods
Cite the treatment (N, etc.) in the text and no in Table 1. There are 8 different treatments and it does not appear to be a combination of them.
The item “Methods for determination of agronomic indicators” is very summarized: Were the data evaluated at harvest? How was the corn harvested? Did they cut it close to the ground? How did they harvest it? Did they pull it out, separate the parts???
I suggest an evaluation of the intercropping's efficiency.

Results
The manuscript contains an excessive number of figures and tables. Try to combine or insert the data into the text. Figures and tables with repeated results should be eliminated. Figures and tables should be self-explanatory. Their titles should include an explanation of N, NxM, etc. Do not use T1, ...T8 in tables and figures. Cite the treatment and explanation in the title. Cite figures and tables in the text.

Discussion
In the Discussion item, the authors basically presented a general review, lacking a discussion of the results obtained.

After all these adjustments, the manuscript has to be re-evaluated.

Experimental design

Methods
Cite the treatment (N, etc.) in the text and no in Table 1. There are 8 different treatments and it does not appear to be a combination of them.
The item “Methods for determination of agronomic indicators” is very summarized: Were the data evaluated at harvest? How was the corn harvested? Did they cut it close to the ground? How did they harvest it? Did they pull it out, separate the parts???
I suggest an evaluation of the intercropping's efficiency.

Validity of the findings

See observations in the manuscript.

Additional comments

I recommend that authors search for Peer J articles to rephrase their manuscript.

---

## Round 0.2 · Minor Revisions

Kindly, address the minor revisions required.

Reviewer 1 ·

Basic reporting

It can be published after the corrections I have made and uploaded to your system are made.

Experimental design

Sufficient

Validity of the findings

Sufficient

Annotated reviews are not available for download in order to protect the identity of reviewers who chose to remain anonymous.

·

Basic reporting

I am satisfied with the current version

Experimental design

I am satisfied with the current version

Validity of the findings

I am satisfied with the current version

·

Basic reporting

No comment

Experimental design

Well structured

Validity of the findings

Intercropping is a very important cultural practice in agriculture, hence the validity of the results obtained. Furthermore, corn cultivation is important for the population of the entire world.

Additional comments

Dear authors:
I commend the authors for their comprehensive response to the requested revisions. All comments have been addressed with considerable detail.
To further enhance the manuscript quality, I recommend careful proofreading to correct grammatical, spacing errors, misconfiguration errors and others, which currently detract from the overall presentation.
A few observed errors are as follows:
. On line 82, correct SIMON (Simon…, Zhou et al. date??)
. On line 85, eliminate space after (Wang et al. 2024a),..
. On line 88 and others, italic in scientific name (Aconitum carmichaeli, Eugenia dysenterica,
. On line 256, insert space (growth. The..)
. On line 463, correct traditional Chinese medicine (Traditional Chinese Medicine)
. On line 560, correct chronological order of authors (Chen et al. 2017; Kiss et al. 2004).
. On line 597 correct signal (of saponins-a key bioactive compound-by reducing…)
. On line 647 change full stop to comma (various indicators. X.C. H D and W.L: various indicators, X.C. H D and W.L)

Sincerely,

---

## Round 0.3 · accepted · Accept

Thanks for addressing the required comments; now the manuscript is ready for publication.

- Authors for two references are given in all caps and this needs to be fixed (SIMON; SU).
- Figure 1 legend, italicize "Ophiopogon japonicus" lowercase "Corn"
- Figure 2 legend lowercase "Parts"